# The Ghost of Predator Past: Interaction of Past Predator Exposure and Resource Availability on Toxin Retention and Cell Growth in a Dinoflagellate

**DOI:** 10.3390/toxins17060290

**Published:** 2025-06-07

**Authors:** Gihong Park, Christina Batoh, Hans G. Dam

**Affiliations:** Department of Marine Sciences, University of Connecticut, 1080 Shennecossett Road, Groton, CT 06340, USA; christina.batoh@gmail.com (C.B.); hans.dam@uconn.edu (H.G.D.)

**Keywords:** *Alexandrium catenella*, paralytic shellfish toxin, past predator-induced toxin, predator legacy effects, non-consumptive effects, toxin retention, toxin maintenance cost, defense-growth tradeoff

## Abstract

The non-consumptive effects of past predator exposure on phytoplankton have gained recognition, but how these effects are modulated by resource availability requires further study. We examined the simultaneous effects of past predator exposure (direct, indirect, and no exposure) and nutrient regime (combinations of N- and P-repletion and limitation) on the paralytic shellfish toxin retention and cell growth rate of a toxic dinoflagellate, *Alexandrium catenella* (strain BF-5), under a laboratory-simulated bloom condition (exponential, stationary, and declining phases). Within a past predator exposure treatment, cell toxin retention was generally higher under N-replete than N-limited conditions. The cells of past direct predator exposure treatment retained or produced more toxin than those in the indirect-exposure or no-exposure treatments regardless of nutrient regime in the exponential and stationary phase. By contrast, cells directly exposed to predators showed lower growth rates than the other two treatments, and also showed a tradeoff between toxin retention rate and growth rate. Separate experiments also showed that the effect of past predator exposure on reducing cell growth is stronger under N repletion than N limitation. These results imply that the interactions of past predator exposure and resource availability impact bloom dynamics and toxin transfer in the food web.

## 1. Introduction

The dinoflagellate genus *Alexandrium*, which produces a suite of potent neurotoxic alkaloids known as paralytic shellfish toxins (PST) [1,2], is a model system for the study of inducible defense in phytoplankton. Predators and predator-related cues (kairomones, or alga-to-alga alarm signals from cell death or destruction via predation) increase PST production in *Alexandrium* species like *A. minutum* [3,4,5,6,7,8,9], *A. tamarense* [10], and *A. catenella* [11,12,13,14,15,16].

Issues that have received less attention, however, relate to whether predator-induced toxin production continues when predators or their cues are no longer present (the ghost of predator past), and how nutrient availability modulates cell toxin production and growth after predator exposure. Non-consumptive predator legacy effects affect prey behavior and population dynamics [17,18,19] and have gained recognition in toxic phytoplankton prey [5,10,20,21]. The time trajectory of inducible defense is faster than the relaxation of defense in terrestrial plants [22,23] and phytoplankton [10,24]. Consistent with this latter notion, bloom predictions of the dinoflagellate *Dinophysis* spp., which produces diarrheic shellfish poisoning (DSP) toxins, improved when both predator biomass and predator cues were considered with a time lag of more than one month [21], suggesting long lingering predator effects.

Predator-induced toxin production can result in a fitness cost as finite resources allocated to defense come at the expense of growth, reproduction, energy storage, or fighting disease [25,26,27]. Thus, the reduction in cell division rate that arises from an increase in toxin production rate represents an allocation cost [16,28], although such costs are not always evident in *Alexandrium* [8,20]. Predator legacy effects may also be costly because prey continue to spend resources to avoid autotoxicity and to store and transport toxins in and out of vacuoles [22,29]. Because the production of inducible defenses requires the allocation of exogenous resources, the nutrient regime may also modulate predator-induced legacy effects on toxin production and retention. Nitrate (N), phosphate (P), and their ratio (N:P) affect the cell growth rate and toxin production of *Alexandrium* [30,31]. Cell toxin content (CTC) is greater under N-replete compared to N-depleted conditions, since STX and its analogues are N-rich molecules; moreover, *Alexandrium* species are characterized by high internal levels of glutamine and arginine (N-rich amino acids) as precursors of STX [32]. P-limitation increases toxin levels of *A. tamarense* and *A. minutum* independently of N-levels [30,33]. In laboratory experiments using *A. minutum*, cell growth rate (CGR) decreased when the N:P ratio deviated either upward or downward from the Redfield ratio (N:P = 16), and CTC increased with both N and P limitation [34]. CTC increases under high N:P ratios in *A. tamarense* [30,32,35] and in *A. minutum* [31].

In the present study, we examined the effect of the interaction of past predator exposure and nutrient regime on the toxin retention and growth of *Alexandrium catenella*. We asked the following: (1) Does past predator exposure (directly or indirectly) affect toxin retention and cell growth? (2) Does toxin retention in the context of past predator exposure depend on nutrient regime? (3) Does toxin retention in the context of past predator exposure incur a cost to growth? We hypothesized that toxin retention may be highest in the past direct predator exposure treatment under the N-replete condition, but cell growth shows the opposite. Thus, there would be a tradeoff between toxin retention and cell growth, as in the case of predator-induced toxin production and growth.

## 2. Results

### 2.1. Assay 1: Cell Growth Rate (CGR), Past Predator Exposure vs. Nitrogen Availability

In this assay, cells were first induced to produce toxins by predators, then their growth rate was monitored after predator removal (treatment). A control consisted of cells that were not induced to produce toxins by predators. The treatment and control cells were simultaneously compared in conditions that were N-replete or N-limited (Appendix A). Natural logarithm-transformed data of cell number were combined for replicates (*n* = 10) within each treatment and plotted versus day. The linear regressions for each treatment have different slopes (ANCOVA, *p* < 0.001, Appendix A), indicating differences in growth rates among the treatments, confirmed by a Kruskal–Wallis one way ANOVA on ranks (*p* < 0.001), with all four treatments differing from each other (post-hoc SNK, *p <* 0.05). Plotting the data as a reaction norm shows the trend in the data and an interaction effect between past predator exposure and N availability on CGR (Figure 1 and Appendix A; two-way ANOVA, *F*_1,36_ = 18, *p* < 0.001). As expected, in the control cells, CGR was higher under N-replete (880 µM L^−1^) than N-limited (3 µM L^−1^) conditions. In the treatment cells, however, the pattern was the opposite. CGR was lower in N-replete than N-limited media. Thus, the decrease in CGR linked to past predator exposure was more pronounced under N repletion than N limitation.

### 2.2. Assay 2: Effect of Past Predator Exposure and Nutrient Regime on CTC and CGR

Like Assay 1, in this assay, cells were first exposed to predators to induce toxin production (direct and indirect induction), and their CTC and CGR were monitored after the predators were removed. Comparisons were made relative to a control (no previous predator exposure). We consider the differences within a given treatment and between treatments as a function of nutrient regimen for the exponential, stationary and declining phases of the *Alexandrium* bloom.

*EP: Exponential Phase.* CTC differed significantly (±1 s.d. of the mean) among the previous predator exposure treatments (past direct predator exposure (74 ± 16 fmol cell^−1^) > past indirect predator exposure (33 ± 7 fmol cell^−1^) > no predator exposure (18 ± 7 fmol cell^−1^): *F*_2,18_ = 440, *p* < 0.001, Post-hoc SNK), the different nutrient regimes (*F*_5,18_ = 20.9, *p* < 0.001), and their different interactions (Figure 2A and Appendix A; two-way ANOVA, *F*_10,18_ = 5.46, *p* < 0.001). Nutrient regime had a significant effect on CTC among the past direct exposure treatments (Figure 2A; M2 ≤ M3 = M1 ≤ M6 = M5 < M4: *F*_5,6_ = 15.2, *p* = 0.002, Post-hoc SNK) and the control (Figure 2A; M2 = M1 < M4 ≤ M3 = M5 ≤ M6: *F*_5,6_ = 40.2, *p* < 0.001, Post-hoc SNK). The post-hoc comparisons of means revealed significant effects of nitrate concentration on CTC in the past direct exposure treatment and the control (3 = 55 < 440 = 110 µM L^−1^ [NO_3_^−^]: *F*_3,8_ = 14.9, *p* = 0.001 and 3 < 110 = 55 < 440 µM L^−1^ [NO_3_^−^]: *F*_3,8_ = 60.6, *p <* 0.001, Post-hoc SNK, respectively) and the N:P ratios in the control (0.1 = 3 < 16 < 440: *F*_3,8_ = 39.2, *p <* 0.001, Post-hoc SNK), but no effect of phosphate (*F*_3,8_ = 0.163, *p =* 0.918). However, the effect of previous predator exposure on cell growth rate (CGR) showed a reversal of the pattern shown by CTC (past direct grazer exposure < past indirect predator exposure = no predator exposure: *F_2_*_,*72*_ = 48.2, *p* < 0.001, Post-hoc SNK), and there was no pattern shown for nutrient regime (*F*_5,72_ = 1.28, *p =* 0.280), but there was an interaction between predator exposure and nutrient regime (Figure 2B and Appendix A; two-way ANOVA, *F*_10,72_ = 2.03, *p =* 0.042). There were, in addition, significant effects on CGR among media in the past indirect exposure treatments (Figure 2B; M5 ≤ M4 = M1 = M6 ≤ M2 = M3: *F*_5,24_ = 4.14, *p* = 0.009, Post-hoc SNK) and the control (Figure 2B; M1 ≤ M2 = M4 ≤ M5 = M3 = M6: *F*_5,24_ = 3.39, *p* = 0.019, Post-hoc SNK). The effects of elevated N:P ratios on CGR were only significant in the control (0.1 = 3 ≤ 16 < 440 (N:P): *F*_3,26_ = 4.91, *p =* 0.008, Post-hoc SNK).

*SP: Stationary Phase.* Similar to the exponential phase, there were significant effects of previous predator exposure on CTC (past direct predator exposure (76 ± 17 fmol cell^−1^) > past indirect predator exposure (48 ± 7 fmol cell^−1^) > control (34 ± 7 fmol cell^−1^): *F*_2,18_ = 372, *p* < 0.001, Post-hoc SNK), the nutrient regime (*F*_5,18_ = 39.3, *p* < 0.001), and the interaction of the two (Figure 3A and Appendix A; two-way ANOVA, *F*_5,18_ = 7.77, *p* < 0.001). CTC differed with nutrient regime in all grazing treatments and the control (Figure 3A; ANOVA, past direct predator exposure (M1 = M2 < M4 = M6 = M3 < M5: *F*_5,6_ = 23.0, *p* < 0.001), past indirect predator exposure (M2 = M1 = M4 < M3 = M5 < M6: *F*_5,6_ = 21.2, *p* < 0.001), and the control (M2 = M1 = M4 ≤ M6 = M5 = M3: *F*_5,6_ = 8.86, *p* = 0.001), Post-hoc SNK, respectively). Specifically, the comparisons revealed significant effects of both nitrate concentrations and N:P ratios on CTC in the past direct exposure group (3 < 110 < 55 = 440 µML^−1^ [NO_3_^−^]: *F*_3,8_ = 7.61, *p* = 0.001 and 0.1 = 3 < 440 = 16 (N:P): *F*_3,8_ = 6.04, *p =* 0.019, Post-hoc SNK), in the past indirect predator exposure group (3 = 110 < 55 < 440 µML^−1^ [NO_3_^−^]: *F*_3,8_ = 15.2, *p* = 0.001 and 3 = 0.1 < 16 < 440 (N:P): *F*_3,8_ = 11.4, *p =* 0.003, Post-hoc SNK), and in the control (3 = 110 < 440 = 55 µML^−1^ [NO_3_^−^]: *F*_3,8_ = 15.2, *p* = 0.001 and no difference in N:P ratio: *F*_3,8_ = 2.84, *p =* 0.106, Post-hoc SNK). Like the pattern shown by the EP, the effect of previous predator exposure on CGR was significant (past direct predator exposure < past indirect predator exposure = control: *F*_2,72_ = 27.5, *p* < 0.001, Post-hoc SNK), while there was no effect of nutrient regime (*F*_5,72_ = 0.57, *p =* 0.726) or the interaction between predator exposure and nutrient regime (Figure 3B and Appendix A; two-way ANOVA, *F*_10,72_ = 0.68, *p =* 0.740).

*DP: Declining Phase.* There were significant effects of predator exposure in the declining phase (past direct predator exposure (36 ± 4 fmol cell^−1^) > past indirect predator exposure (30 ± 9 fmol cell^−1^) > control (24 ± 5 fmol cell^−1^): *F*_2,18_ = 27.3, *p* < 0.001, Post-hoc SNK) and the interaction (two-way ANOVA, *F*_5,18_ = 7.94, *p* < 0.001), but no significant difference in the nutrient regime (Figure 4A and Appendix A; *F*_5,18_ = 1.91, *p =* 0.143). Although there were wide fluctuations, CTC significantly differed with the medium M4 in the indirect treatment (Figure 4A; M2 = M6 = M1 = M5 = M3 < M4: *F*_5,6_ = 10.7, *p =* 0.006, Post-hoc SNK). However, similar to the results for the SP, previous predator exposure had a significant effect on CGR (Figure 4B; past direct predator exposure = past indirect predator exposure < control: *F*_2,69_ = 9.26, *p* < 0.001, Post-hoc SNK), but there was no effect of either nutrient regime alone, or an interaction effect (Figure 4B and Appendix A; *F*_5,69_ = 0.74, *p* = 0.594 and two-way ANOVA, *F*_10,69_ = 0.75, *p* = 0.673, respectively).

*All Phases.* Regardless of growth phase, past predator-exposed cells showed significantly higher CTC (past direct predator exposure (62 ± 23 fmol cell^−1^) > past indirect predator exposure (37 ± 11 fmol cell^−1^) > no predator exposure (26 ± 9 fmol cell^−1^): *F*_2,105_ = 49.8, *p* < 0.001, Post-hoc SNK). Significantly higher CTCs were observed for N-replete media and elevated N:P ratios in the past direct predator exposure treatment and the control cells from EP and SP. CGR showed, however, a reversed pattern to CTC (past direct predator exposure < past indirect predator exposure = control: *F*_2,264_ = 46.2, *p* < 0.001, Post-hoc SNK), indicating a possible tradeoff between toxin retention and cell growth.

### 2.3. Relationships Between Toxin Retention Rate and Cell Growth Rate

There was a significantly negative relationship between PST retention rate (linear or exponential) and growth rate, and PST retention rates given by linear and exponential models with CGR showed no difference when the regression slopes were compared (ANCOVA, *p* = 0.345; linear: y = 0.1 − 1.54x and exponential: y = 0.1 − 1.49x), but a higher *r^2^* was shown for the linear model (Figure 5A and Appendix A; *r^2^_linear_* = 0.53 and *r^2^_exponential_* = 0.42, *p* < 0.001, *n* = 54). Hence, for parsimony, we used the linear model in this assay. Post-hoc comparisons of means revealed significant differences in PST retention rate among past grazer exposure treatments relative to the past indirect grazer exposure treatments and controls (past direct grazer exposure > past indirect grazer exposure = no grazer exposure: *F*_2,51_ = 17.3, *p* < 0.001, Post-hoc SNK). Furthermore, tradeoffs were more evident in exponential and stationary growth phases (Appendix A; EP, *r^2^* = 0.70, *p* < 0.001, *n* = 18; SP, *r^2^* = 0.60, *p* < 0.001, *n* = 18; and DP, *r^2^* = 0.54, *p* = 0.002, *n* = 18).

The tradeoff between PST retention rate and growth rate can also be explored by considering the PST maintenance cost. This cost relates to the reduction in growth rate of the treatment (past predator exposure) relative to the control (no predator exposure) [15,36,37]. We calculated relative PST (PST_treatment_ − PST_control_) [15,38,39], and all data were normalized by Z-scores (standardization). The maintenance cost was higher when cells were previously exposed directly to predators, compared to indirectly (Figure 5B; *t*-test, *t*_17,17_ = 4.67, *p <* 0.001, *n* = 36). There was also a difference in relative PST content between cells that had previously been exposed to predators directly versus indirectly (Figure 5B; *t*-test, *t*_17,17_ = 4.46, *p* < 0.001, *n* = 36).

## 3. Discussion

As we hypothesized, the cell toxin content (CTC) was different in cells previously exposed (directly or indirectly) to predators compared to cells not exposed to predators, and the difference depended on nutrient availability or growth phase, suggesting the differential plasticity of toxin retention during a bloom. In general, the CTC in some *Alexandrium* species is typically highest in the exponential phase, when cells are growing fast [30,32,40], whereas others have shown an increase in the stationary phase [15,41,42]. We also observed that the constitutive PST was highest in the stationary phase among all controls. Furthermore, similar to direct contemporaneous exposure to predators [15], there were significant legacy effects of past predator exposure on CTC (past direct predator exposure > past indirect predator exposure > no predator exposure). This result is consistent with the hypothesis that direct exposure to predators has a greater effect on toxin production than feeding-related cues such as kairomones [43] or algal alarm cues [44]. Our results also reveal that the legacy effect in toxin retention may last for generations after predator exposure.

The relationship between CTC and cell growth rate (CGR) in *Alexandrium catenella* is summarized in John and Flynn [45]: the N limitation of toxin synthesis is evident when cell toxin content increases after the N refeeding of cultures. Other *Alexandrium* species showed asymptotical increases in growth rate when increasing nitrogen concentration toward a maximum [46,47,48] with growth limited below 10 μM [NO_3_^−^]. In contrast, because P limitation reduces cell growth, but not PST synthesis, P limitation increases cell toxin content [30,31,32,33]. Thus, CTC and CGR are coupled under N limitation, but not P limitation. Selander et al. [4] tested for predator-induced PST production under different nutrient regimes and documented a significant indirect induction of PSP production in N-replete conditions. The same phenomenon was observed in our assay on the interaction of previous predator exposure and nutrient regime. When offered N-replete media either in M3, M4, and M5 (balanced N:P ratio: 16) or in M6 (unbalanced N:P ratio: 440), a significant increase in the CTC of cells directly and indirectly exposed to predator cues was observed in the EP and SP.

These observations also indicate that CTC is also controlled by N:P ratio. In the EP and SP experiments, the highest CTC of control cells was usually observed when the ratio was 440:1 (M6 medium), consistent with previous works [32,33,49,50]. Moreover, dual nutrient limitation (M2; 3:1) resulted in the lowest CTC level among media, which is possibly explained by the rapid deregulation of C metabolism associated with C-specific growth determination and C-specific toxin content [31]. Overall, the N-source primarily impacts toxin production, while P limitation has the potential to affect growth, and shows numerous insidious effects by disturbing biochemical regulations such as the phosphorylation of intermediates, C fixation, and toxin synthesis [31]. Hence, in the present study, CTC regulation induced by N limitation was more evident than that induced by P limitation in all treatments, except in the declining phase (DP). More importantly, even though an effect of nutrient regulation on CTC was evident, it was less than the effect of predator-regulated CTC. The partial eta squared (*η^2^*) was used to evaluate how large of an effect the independent variables had on the dependent variable. Thus, the effect of both past predator exposure and nutrient regime on CTC observed here are great (effect size for group mean differences: *η^2^*, past predator exposure versus nutrient regime, 0.98 vs. 0.85 in EP and 0.98 vs. 0.92 in SP), except for 0.75 vs. 0.35 in DP.

The pattern for CGR was, however, opposite to that shown by CTC—the significant decreases in growth rate were pronounced as a result of the effect of past predator exposure relative to controls in the EP and SP (past direct predator exposure < past indirect predator exposure = no predator exposure) and in the DP (past direct predator exposure = past indirect predator exposure < no predator exposure). Interestingly, the CGR of cells in past direct predator exposure treatments in all phases decreased to zero, where the cell division rate could not exceed the cell mortality rate even in the N- and P-replete media. There were, in addition, no significant effects of media on growth rates in either different treatments or different interactions. The effect sizes for group mean differences on CGR confirmed a greater effect of past predator exposure than nutrient regime in all growth phases (*η^2^*, past predator exposure versus nutrient regime, 0.57 vs. 0.08 in EP; 0.43 vs. 0.04 in SP; and 0.21 vs. 0.05 in DP).

Because the energy gained from resources cannot be maximized among all biological functions simultaneously [25,26,51], a negative relationship between the PST retention rate and cell growth rate implies a defense–growth tradeoff, leading to a cost associated with maintaining past predator-induced PST. Selander et al. [4] examined the effects of nitrate on predator-induced toxin production in *A. minutum,* and found a significant increase in CTC in N-replete treatments, but no detectable reduction in CGR, suggesting that the tradeoff was negligible in that species. However, there was evidence of a tradeoff between toxin production rate and growth rate in *A. catenella* [14,15]. These defense–growth tradeoffs were studied under nutrient-replete conditions in the exponential phase. Yet, the ecological consequences of this tradeoff may be more pronounced under N limitation if *Alexandrium* cell disproportionately allocates more nitrogen to produce N-rich PSP toxins to increase the grazing defense [52]. Alternatively, cells may devote all resources to growth under extremely low nutrient conditions, and forego inducible toxin production, and thus grazing defense, leading to different ecological consequences.

Two classes of chemical defense costs affect prey resource allocation—the direct costs of defense production and maintenance [22,29,53]. Recent work on dinoflagellates has addressed the direct fitness cost of PST production [14,15]. Park et al. [15] found significant differences in fitness reduction (direct fitness cost) associated with predator-induced toxin production between constitutive and inducible defenses using a quadrant plot. In this study, a similar approach showed that toxin maintenance in the context of past predator exposure incurs a fitness reduction cost (Figure 5B). This cost is evident regardless of the various nutrient regimes, suggesting strong legacy effects of predators on prey toxin production and growth. Assay 1 also revealed a maintenance cost of defense using *Alexandrium* cells that had previously been indirectly exposed to predators (similar predator-to-cell ratios to assay 2). The relative maintenance cost was higher under conditions of N repletion than N limitation, implying the cells retaining higher PST in the N-replete media may continue to maintain the levels of toxisn as the nutrients required for maintenance are in abundant supply [4]. The cells induced in N-limited media, however, may not have an adequate supply of nutrient resources to maintain the induced levels of PST, and preferentially allocate nutrients and energy to cellular growth.

The hypothesis of “the ghost of predation past” [54,55] posits that a species subject to past selection for anti-predator behavior will retain the behavior if it is not too costly [55], as cited by Blumstein et al. [56]. For example, infochemicals from rotifers continued to induce colony formation in the chlorophyte *Scenedesmus obliquus* after the rotifers were removed [24]. A slow relaxation of defenses once that selective pressure is removed may occur because many types of morphological defenses do not require high maintenance cost [24,57]. Slow chemical defense relaxation is evident in *A. tamarense* [10], even as the cost of defense is not trivial [16]. Defense induction and relaxation, however, should be framed in terms of the net balance of fitness cost and benefits. Generally, the maintenance cost of retaining PST is linked to toxin storage and minimizing autotoxicity, in which context the cell voltage-gated ion channels must be protected from the toxins the prey produce [58,59]. Although many terrestrial plants infiltrate their own tissues with nerve toxins that are relatively nontoxic to themselves because they lack a nervous system but are broadly toxic to herbivores, plants including toxigenic phytoplankton are still confronted with the problem of autotoxicity [60], as well as space for storing toxins in the cell’s vacuole and the machinery for transporting the compounds into/out of the vacuoles [22,29]. The cost of toxin maintenance is evident in the tradeoff we observed, in which toxin retention comes at the cost of reduced growth. The benefit of retaining the toxin in an ecology of fear relates to predation deterrence [61,62], with the balance of cost and benefit increasing as a function of predator density [16]. Future studies could experimentally test this hypothesis by looking at both defense induction and relaxation in the context of different kind of predators and their densities.

Ecological consequences: Harmful algal blooms are complex phenomena whose dynamics depend both on resource availability and predation. Blooms of toxic prey add a layer of complexity because of the induced defense (toxin production) that comes at the cost of a reduced cell division rate. If prey retain toxins once predation pressure decreases, nutrients are depleted faster and cell growth decreases, thus potentially shortening the duration of the bloom. Furthermore, maintaining high toxin production after the predator has gone is maladaptive (keeps the prey from its fitness peak). But slow defense relaxation may have evolved in environments in which the threat of predation fluctuates. In such an environment, it would pay to have fast defense induction and slow defense relaxation.

## 4. Conclusions

The main goal of this study was to examine the combined top-down (predator-related cues) and bottom-up (nutrient regime) influences on the control of past predator-induced PST and cell growth in the toxic dinoflagellate *Alexandrium catenella.* Four main outcomes were apparent. Firstly, past predator exposure had a strong effect on the maintenance/retention of PST in the exponential and stationary phases. Secondly, regardless of nutrient regime, cells induced to produce toxins in response to previous direct exposure to predators showed lower growth rates compared to cells that had not been exposed to predators. Thirdly, these results suggest that maintaining past predator-induced PST comes at the expense of cell growth, and that this tradeoff may last for generations after predator exposure. Lastly, the cost of PST retention is constrained by both predator exposure and the nutrient regime, and the effect of the former is more pronounced. Thus, this legacy of past grazers may shorten bloom duration.

## 5. Materials and Methods

### 5.1. Experimental Design

We used an approach similar to those in previouss work in our lab [11,15], which focused on predator-induced toxin production. In those experiments, the cell toxin content (CTC) and cell growth rate (CGR) of *Alexandrium catenella* strain BF-5 were measured in 72 h incubations in treatments in which cells (i) were not separated from copepod predators (direct exposure), (ii) separated from the predators by a mesh (indirect exposure), or (iii) not exposed at all to predators or their cues (control). In the present study, cells from the same *A. catenella* were first induced to produce toxins by exposure to predators or their cues; then their CTC, CGR, or both, depending on the experiment, were applied in the absence of predators or their cues for several cell divisions and compared to controls (no previous predators or predator cue exposure).

### 5.2. Culturing of Prey and the Zooplankton Predator

*Alexandrium catenella* was isolated from the Bay of Fundy, Canada, and grown in semi-continuous cultures in f/2 medium without silicate [63] using 0.2 µm filtered seawater from Long Island Sound. Seed cultures were kept in the exponential growth phase in an 18 °C environmental chamber illuminated with fluorescent lighting (100 µM m^−2^ s^−1^) set to a 12 h:12 h light:dark photoperiod. The strain (known as BF-5 or NB-05) maintained its constitutive toxin content (~30 fmol cell^−1^) in our laboratory for two decades [64,65].

A population of the predator copepod *Acartia hudsonica* was collected from Casco Bay, ME, USA. Here, blooms of toxic *A. catenella* are common, and *A. hudsonica* is known to feed upon the toxic cells [66,67]. Copepod cultures were maintained for several generations prior to the assays in an environmental chamber set to the same conditions as described above for the cultures of *A. catenella*. Copepods were fed a non-limiting, mixed diet of *Thalassiosira weissflogii*, *Tetraselmis* sp., and *Rhodomonas* sp., and were also cultured under the same conditions as *A. catenella,* except that the diatom, *T. weissflogii*, was grown in f/2 medium containing silicate.

### 5.3. Assay 1: Cell Growth Rate (CGR), Past Predator Exposure vs. Nitrogen Availability

In this experiment, cells were first indirectly induced to produce toxins by exposing them to predator cues for 72 h. These exposures increased their CTC by 53% relative to control cells [68]. Induced cells from one replicate beaker were collected on a 10 µm mesh and repeatedly washed with filtered sea water (FSW). Approximately half of the induced cells were reconstituted in 250 mL of N-replete, f/2 medium (880 µM L^−1^ [NO_3_^−^]; 36.3 µM L^−1^ [PO_4_^3−^]). The remaining induced cells were diluted into 250 mL of N-limited media (3 µM L^−1^ [NO_3_^−^]; 1 µM L^−1^ [PO_4_^3−^]) in 0.2 µm FSW from Long Island Sound. The latter significantly limits the growth of *Alexandrium* species [41,69]. Control cells that had not been induced were handled in the same manner as induced cells and split into the N-replete and N-limited media as above. Thus, we had induced treatments and controls under N repletion and N limitation. Cells were incubated in 50 mL glass tests tubes at a concentration of 25 cells mL^−1^ (*n* = 10). The tubes were lightly capped and kept in an environmental chamber set to 18 °C and a 12 h:12 h light:dark cycle. Over a period of 11 days, 1 mL subsamples were taken from each tube every 2 days, and preserved in acidic Lugol’s solution. The subsamples were preserved in individual wells of 24-well culture plates. The cells were enumerated using an inverted microscope. Growth rate was estimated assuming exponential growth from the slope of the Ln-transformed cell concentrations vs. time.

### 5.4. Assay 2

#### 5.4.1. CGR, Effect of Past Predator Exposure and Nutrient Regime

Cells were separated from the copepods and their cues (Assay 2, Figure 6B) and carefully washed six times on a 10 µm wet sieve, before being re-inoculated into six separate media (labeled M1 through M6; M2 is 0.2 µm FSW). Vitamins and trace metals were added at concentrations equivalent to f/2 media into the filtered seawater, but with the exception of the M2 media treatment, the concentrations of NO_3_^−^ and PO_4_^3−^ were manipulated in combinations with [NO_3_^-^] ranging from 3 to 440 µM L^−1^ and [PO_4_^3−^] ranging from 1 to 28 µM L^−1^, indicating that media were either limited in N, limited in P, limited in both N and P, or replete in both N and P. The nutrient combinations represent N:P ratios between 0.1 and 440 (Assay 2, Figure 6B). We did not analyze nutrients at the ends of assays, but we measured initial N and P (3 µM L^−1^ [NO_3_^−^]; 1 µM L^−1^ [PO_4_^3−^]) using a nutrient analyzer (SmartChem^®^ 200, AMS Alliance, Weston, FL, USA) to set up exact media concentrations.

Cells were incubated in 50 mL sterilized screw-top polycarbonate culture tubes kept under the same temperature and light as in the culturing conditions (Section 5.2). Five replicates for each media treatment were prepared with cell concentrations starting at 25 cells mL^−1^. Over a period of 15 days, 1 mL subsamples were taken from each tube every fourth day and preserved in 0.5% acid Lugol’s solution for cell counting under an inverted microscope (Olympus IX70 Model). A subsample of at least 150 cells was counted on fields of 2.2 or 1.1 mm (22 mm eyepiece magnification) in a well plate.

#### 5.4.2. CTC, Effect of Past Predator Exposure and Nutrient Regime

Park et al. [15] measured initial cell toxin contents (CTCs) in direct and indirect copepod exposure treatments and in controls at the exponential (ES), stationary (SP) and declining (DP) phases of an *A. catenella* laboratory bloom (Figure 6A). After the 72 h incubations, cells directly exposed to copepods increased their CTC to 271% (EP), 109% (SP), and 19% (DP). Increases shown by the indirectly exposed cells were to 39% (EP), 18% (SP), and −20% (DP).

Duplicate samples for CTC were taken from all treatments and controls at the end of assay 2, and then cells were pelletized by centrifugation and the pellets were frozen with 0.1 M acetic acid at −80 °C for toxin analysis. To measure CTC, the frozen cells used for toxin analysis were thawed at room temperature and homogenized using a sonic dismembrator on ice (Model 50, Fisher Scientific, Waltham, MA, USA). The complete disruption of the cells was confirmed by microscopic examination, and the lysed cells were filtered through 0.45 mm ultra-filtration centrifuge cartridges. Saxitoxin congeners (gonyautoxins 1 through 4 [GTX1-4], saxitoxin [STX], neosaxitoxin [NEO], C1, and C2) were identified via reverse-phase ion-pairing high-performance liquid chromatography (HPLC) with post-column oxidative fluorescence as described by Oshima et al. [70]. The chromatogram peaks produced from the fluorescent toxin derivatives were integrated for area using Empower^TM^ 2 software. The CTC was quantified by comparing sample peak areas to the peak areas derived from certified toxin standards purchased from the National Research Council(NRC), Canada. The overall CTC is expressed as femto mole unit per cell (fmol cell^−1^).

### 5.5. Relationships Between Toxin Retention Rate and Growth Rate

Earlier work showed a tradeoff between predator-induced toxin production and cell growth rate in *Alexandrium catenella* (strain BF-5: used in [14,15]). To further explore the implications of a tradeoff, we tracked the relationship between PST retention rate and CGR after predators were removed. We expected that once cells induced to produce toxin by predators were removed from the presence of predators or their cues, they would reduce their CTC thought time, as previously observed [10]. Thus, we estimated the PST depuration from the difference in CTC values between the beginning and end of the 15-day experimental incubation. The PST depuration rate, *D_rate_* (d^−1^), was determined linearly by(1)$$-\frac{{PST}_{(t15)\;}-{PST}_{\left(t0\right)}}{\Delta t}$$
and exponentially by(2)$$-\frac{\ln{PST}_{\left(t15\right)}\;-\ln{PST}_{\left(t0\right)}}{\Delta t}.$$

In these equations, *PST*_(*t*15)_ is the CTC of treatments and controls at the conclusion of exposure time (*t* = 15), and *PST*_(*t*0)_ represents the initial CTC at the beginning of exposure time (*t* = 0) of each phase, with Δ*t* being the elapsed time, namely, 15 days. Using the trapezoidal rule to approximate the definite integral, the PST depuration area (*D_area_*) and retention area (*R_area_*) can be calculated and then defined as *R_area_* = *Total PST_area_* − *D_area_*. Thus, the PST retention rate,*R_rate_* (d^−1^) = *R_area_*″,(3)
was calculated by differentiation as(4)$$R_{\mathrm{rate}}=\frac{\int_{0}^{15}[1-\int_{0}^{15}\left(D_{rate}\right)\;dt]dt}{{\Delta t}^{2}}$$

### 5.6. Statistical Analysis

All statistical analyses were performed using SigmaPlot version 11.0 or SPSS version 29 software. After assumption checks for normal distribution and heteroscedasticity for all data, for assay 1, the data did not meet the assumption of normal data distribution for a parametric ANOVA. Instead, we applied the nonparametric Kruskal–Wallis ANOVA on ranks, and all pairwise comparisons among treatments used the Student–Newman–Keuls (SNK) post-hoc procedure. For assay 2, two-way ANOVA was used to test for the effects of previous predator exposure (past direct predator exposure, past indirect predator exposure, and no predator exposure), nitrogen availability (N repletion and N limitation) and nutrient regime (M1 to M6 media), as well as their interactions, on CTC and CGR in each growth phase (EP, SP, and DP, respectively). Pairwise comparisons among treatments were assessed using the SNK post-hoc procedure. A linear regression tested for the tradeoff between PST retention rate (d^−1^) and cell growth rate (d^−1^), and a slope comparison between PST retention rates (linear vs. exponential) was performed with ANCOVA. A T-test was used to confirm differences in relative PST and its maintenance cost between cells previously exposed directly and indirectly to predators.

## Figures and Tables

**Figure 1 toxins-17-00290-f001:**
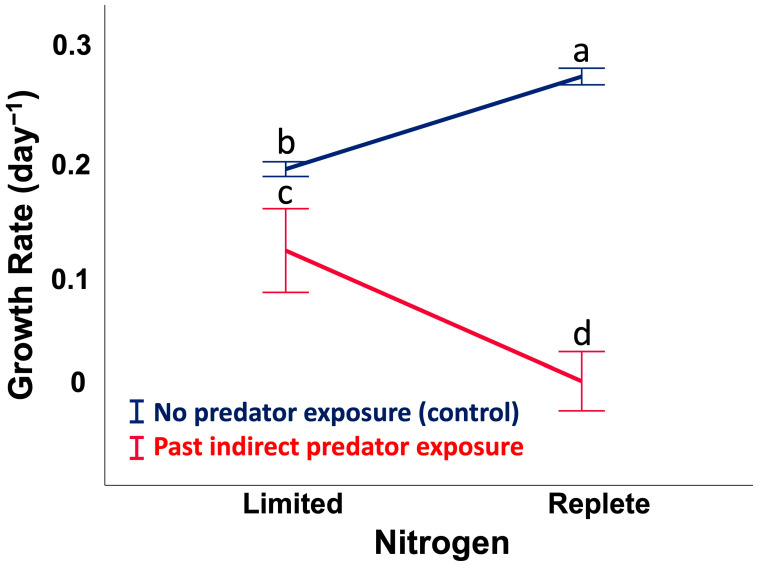
A reaction norm shows the effects of past grazer exposure and nitrogen availability on cell growth rate. N-limited (3 µM L^−1^) indicates 0.2 µm FSW and N-replete (880 µM L^−1^) is f/2 medium. Letters indicate significant differences among treatment means (*p* < 0.05; ANOVA, Post-hoc SNK). Error bars represent ± 1 SD of the mean. The lines are the regression slopes.

**Figure 2 toxins-17-00290-f002:**
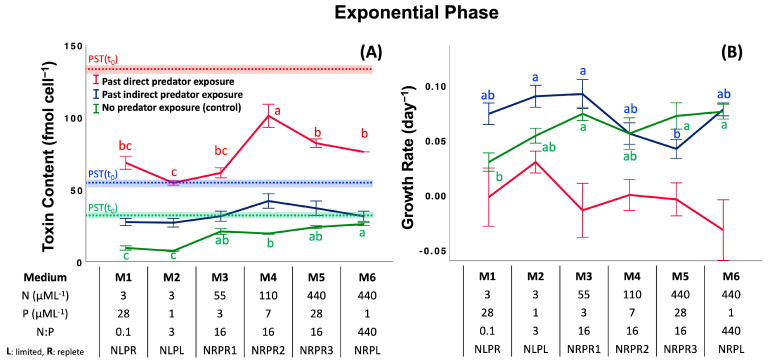
Effect of previous grazer experience and nutrient media on cell toxin content (**A**) and cell growth (**B**) in the exponential phase (EP). Table indicates nutrient concentrations (nitrate and phosphate) and nutrient ratios of media used in the assay. Dotted lines with shaded areas indicate the initial CTC ± 1 standard deviation of treatments and controls at the beginning of exposure time (*t* = 0). Letters represent significant differences among treatment means (*p* < 0.05; ANOVA, Post-hoc SNK). Error bars represent ± 1 SD of the mean.

**Figure 3 toxins-17-00290-f003:**
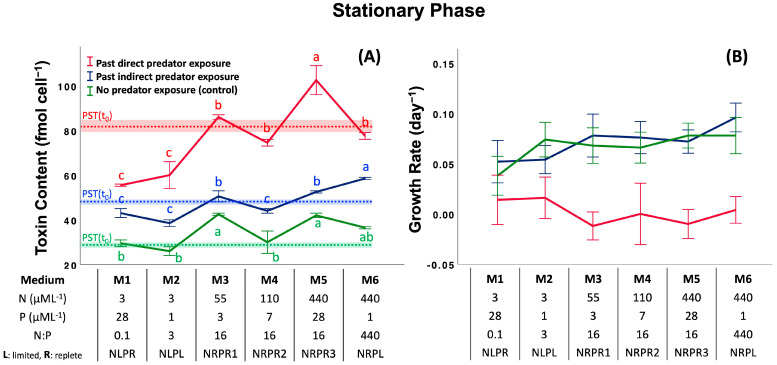
Effect of previous grazer experience and nutrient media on cell toxin content (**A**) and cell growth rate (**B**) in the stationary phase (SP). The table shows nutrient concentrations (nitrate and phosphate) and nutrient ratios of media used in the assay. The dotted lines with a shaded area indicate the initial CTC ± 1 standard deviation of treatments and controls at the beginning of exposure time (*t* = 0). The letters represent significant differences among treatment means (*p* < 0.05; ANOVA, Post-hoc SNK). Error bars represent ± 1 SD of the mean.

**Figure 4 toxins-17-00290-f004:**
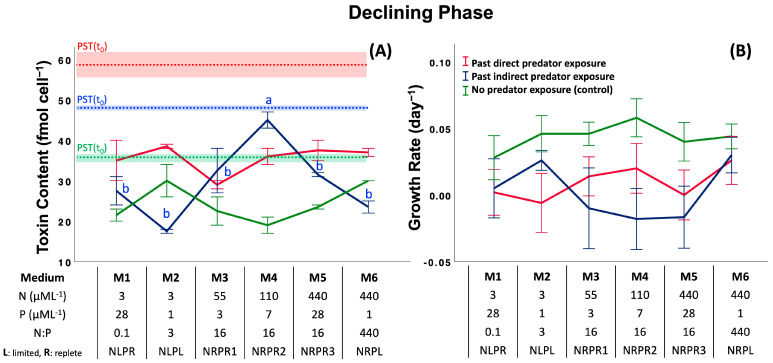
Effects of previous grazer exposure and nutrient media on cell toxin content (**A**) and cell growth rate (**B**) in the declining phase (DP). The table shows the nutrient concentrations (nitrate and phosphate) and nutrient ratios of media used in the assay. Dotted lines with shaded areas indicate the initial CTC ± 1 standard deviation of treatments and controls at the beginning of exposure time (*t* = 0). Letters represent significant differences among treatment means (*p* < 0.05; ANOVA, Post-hoc SNK). Error bars represent ± 1 SD of the mean.

**Figure 5 toxins-17-00290-f005:**
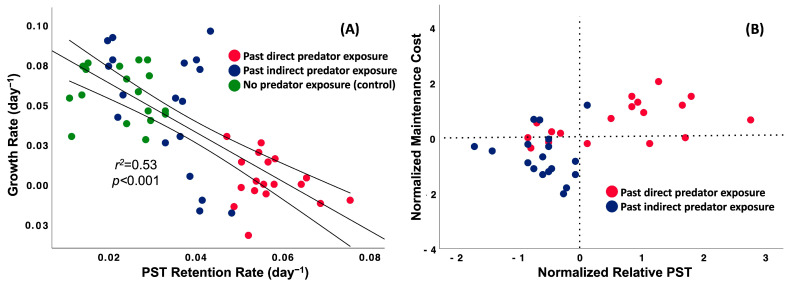
A tradeoff between PST retention rate and cell growth rate (**A**). The lines represent the best fit regression of PST retention rate versus growth rate with 95% confidence intervals. Relationship between the normalized relative PST (*PST_treatment_* − *PST_control_*) and the maintenance cost of past grazer-induced PST (*CGR_control_* − *CGR_treatment_*) as a function of predation type (**B**).

**Figure 6 toxins-17-00290-f006:**
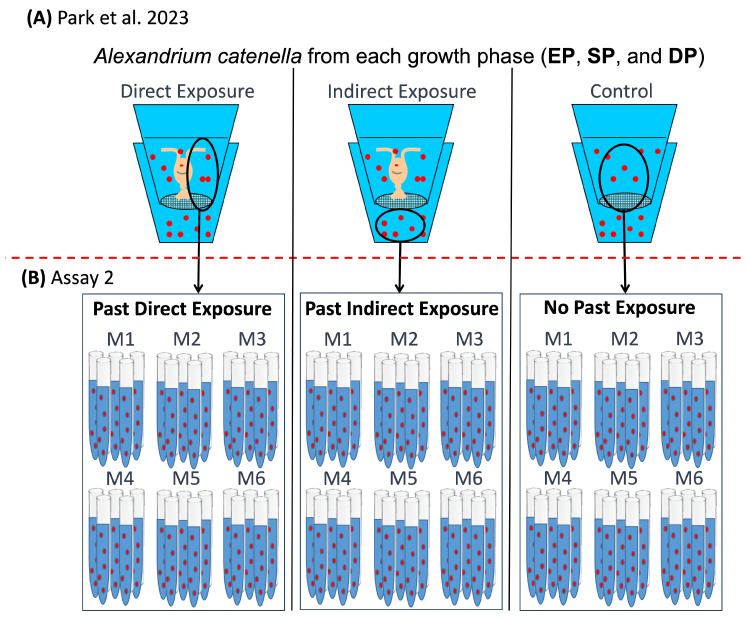
Schematic diagram of experimental design. (**A**) Grazer-induced toxin production by Park et al. [15] in which *Alexandrium catenella* cells were exposed directly to grazers or indirectly to grazer cues. Control cells were not exposed to grazers or cues. The exposure duration was 72 h throughout a simulated bloom (EP, exponential phase; SP, stationary phase; and DP, declining phase). (**B**) An experimental design of assay 2 to test the effects of past grazer exposure and nutrient regime (M1 to M6) on cell toxin content and cell growth rate over 15 days. A red dashed line separates assay 2 from the induction set up designed by Park et al. [15].

## Data Availability

The original contributions presented in this study are included in this article and the Appendix A. Further inquiries can be directed to the corresponding author.

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
