# Peer review of "The Ghost of Predator Past: Interaction of Past Predator Exposure and Resource Availability on Toxin Retention and Cell Growth in a Dinoflagellate"

_toxins, 2025, doi:10.3390/toxins17060290_

Round 1

Reviewer 1 Report

Comments and Suggestions for Authors

Review of the ms. toxins-3607482 - The Ghost of Predator Past: Interaction of Past Predator Exposure and Resource Availability on Toxin Retention and Cell Growth in a Dinoflagellate

The manuscript investigated an intriguing ecological topic in chemical ecology, the influence of  past predator exposure and nutrient regime on dinoflagellate toxin production. It is also aimed at discriminating direct versus indirect predator exposure. The Introduction is very well written, concise, depicting the relevant literature and illustrating the rationale and the working hypotheses.

I only have one main comment on Assay 1. It is not very clear to me why performing this test separately, given that a similar treatment is also included in Assay 2 (past indirect copepod exposure and measurement of CGR in nutrient limited and replete conditions). Moreover, aren’t the results of Assay 1 counterintuitive? Why are the pre-exposed algae reducing the CGR in N-replete conditions compared to N-limited? One could assume that if N is allocated from cell growth to toxin production, then nitrogen limitation should induce even lower CGR. Moreover, the results of Assay 1 do not seem to agree with the results of Assay 2 showed in Fig. 2B, for past indirect exposure and no exposure.

Materials and methods

Line 389: 5.4. Assay 2: Effect of Past Predator Exposure and Nutrient Regime on CTC and GGR. I suggest stating more clearly that the first part of the assay was performed in the previous study Park et al., 2023. For example, in line 395 when the description of Assay 2 in the present study starts.

Line 397: ‘….labeled M1 through M6; M2 was 0.2 μm-FSW).’ If I understood correctly, the M2 was the 0.2 μm-FSW without nutrient addition? It would be useful to indicate this in the figure table.

Results

Figure 1. Why add the lines to connect CGRs in the figure? There are no letters, as written in the legend.

Author Response

We thank the reviewer for their thoughtful concerns, comments, and suggestions, which we address below. Reviewers’ text is in black, and our answers are in red.

The manuscript investigated an intriguing ecological topic in chemical ecology, the influence of  past predator exposure and nutrient regime on dinoflagellate toxin production. It is also aimed at discriminating direct versus indirect predator exposure. The Introduction is very well written, concise, depicting the relevant literature and illustrating the rationale and the working hypotheses. I only have one main comment on Assay 1. It is not very clear to me why performing this test separately, given that a similar treatment is also included in Assay 2 (past indirect copepod exposure and measurement of CGR in nutrient limited and replete conditions). Moreover, aren’t the results of Assay 1 counterintuitive? Why are the pre-exposed algae reducing the CGR in N-replete conditions compared to N-limited? One could assume that if N is allocated from cell growth to toxin production, then nitrogen limitation should induce even lower CGR. Moreover, the results of Assay 1 do not seem to agree with the results of Assay 2 showed in Fig. 2B, for past indirect exposure and no exposure.

Answer: We performed Assay 1 first with past indirectly induced cells, but did not analyze cell toxin contents (CTC). Then, Assay 2 was followed up with past directly and indirectly induced cells and cultured under the various nutrient regime. Although both assays were tested in similar conditions, CGRs of Assay 1 (-0.03 to 0.28 d-1) are higher than Assay 2 (-0.03 to 0.09 d-1, exponential phase). We think the difference in CGR between assays resulted from use of glass tests tubes for Assay 1 and polycarbonate culture tubes for Assay 2 (which has relatively low light transmittance) because CGR of controls in Assay 1 (0.2 to 0.3 d-1) are significantly higher than Assay 2 (0.03 to 0.08 d-1, EP controls). Due to the higher growth rate, the maintenance cost of the retaining PST in Assay 1 is apparent even the assay was tested with past indirectly predator induced cells.

As the reviewer assumed that N limitation should induce even lower CGR, low CGR may be more pronounced under N-limitation if Alexandriumdisproportionately allocates more nitrogen to produce N-rich PSP toxins to increase grazing defense [53]. Alternatively, cells may devote all resources to growth under extreme low nutrient conditions and forego inducible toxin production, and thus grazing defense, leading to different ecological consequences (lines 288-293). In this study, however, the maintenance cost of the relative retaining PST was higher under the N-repletion than N-limitation, implying the cells retaining higher PST in the N-replete media may continue to keep the levels of toxin as the nutrients required for maintenance are in abundant supply [4]. The induced cells in N-limited media, however, may not have an adequate supply of nutrient resources to maintain induced levels of PST and preferentially allocate nutrients and energy to cellular growth (lines 304-309). Additionally, the maintenance cost of retaining PST is linked to minimizing autotoxicity and toxin-stored space in the cell that the vacuole that occupies the induced toxin and the machinery for transporting the compounds into/out of the vacuoles [23,30]. The cost of toxin maintenance is evident in the tradeoff we observed in which toxin retention comes at the cost of reduced growth (lines 325-327).

We agree that the CGR of Assay 1 do not seem to agree with the results of Assay 2 showed in Fig. 2B (due to the use of different test tubes). However, the pattern of cell toxin retention in EP and SP was apparently higher under N-replete than N-limited conditions, so there was no problem that we could show an evidence of defense-growth tradeoff between toxin retention rate and growth rate.         

Materials and methods

Line 389: 5.4. Assay 2: Effect of Past Predator Exposure and Nutrient Regime on CTC and GGR.I suggest stating more clearly that the first part of the assay was performed in the previous study Park et al., 2023. For example, in line 395 when the description of Assay 2 in the present study starts.

Answer: We restated. “As in the previous study of Park et al. (2023), direct and indirect toxin induction were tested simultaneously using experimental cages which are 1 L polycarbonate beakers with nested cages (500 ml) whose bottoms consisted of 10 µm mesh. In brief, A. catenella (initial cell concentration: 300 cells ml−1) within the cage experienced direct toxin induction with 18 adult-female A. hudsonica, whereas cells below the mesh experienced indirect induction (predator cues and the release of prey cellular contents via sloppy feeding). Similar caged containers with no predators served as controls (Assay 2, Fig. 6A).” (lines 406-413 and 443-445)

Line 397: ‘….labeled M1 through M6; M2 was 0.2 μm-FSW).’ If I understood correctly, the M2 was the 0.2 μm-FSW without nutrient addition? It would be useful to indicate this in the figure table.

Answer: Correct. We indicated M2 is 0.2 μm-FSW in the figure table and other figures (Fig.2-4) as well.

Results

Figure 1. Why add the lines to connect CGRs in the figure? There are no letters, as written in the legend.

Answer: It is a reaction norm showing the regression slopes, which indicate the trend in the data and an interaction effect between past predator exposure and N-availability on CGR and we added N concentration for the reader’s information. The letters indicating significant differences are shown now. (lines 89-90 and 99)

Thank you!

Reviewer 2 Report

Comments and Suggestions for Authors

This is an interesting manuscript on the effect of a copepod predator on toxin production and growth rate in a dinoflagellate prey. The MS is well presented. I have a few comments-

Line 16 Space between- the directly

Results; Is it justified to join points indicating different treatments. I suggest that the points are not joined or that bar diagrams are used.

Methods: How many copepods were used per treatment; males or females; gravid or non-gravid adults?

In the direct predation treatments what percentage of the dinoflagellates were consumed?

Author Response

We thank the reviewer for their thoughtful concerns, comments, and suggestions, which we address below. Reviewer’s text is in black, and our answers are in red.

This is an interesting manuscript on the effect of a copepod predator on toxin production and growth rate in a dinoflagellate prey. The MS is well presented. I have a few comments-

Line 16 Space between- the directly

Answer: Fixed. (line 16)

Results; Is it justified to join points indicating different treatments. I suggest that the points are not joined or that bar diagrams are used.

Answer: Yes, we tried to use the bar diagram for the result figures, but  that did not improve the presentation. We thought the line graphs were better to emphasize the pattern for CGR, which is opposite to CTC, between treatments regardless of nutrient regime in the exponential and stationary phase. The M1-M6 media on x-axis are in order of N concentration, so the line to join the points are reasonable. Please bear with us. Instead, we added more information explaining the media in the results and figures (Fig. 2-4 and 6).

Methods: How many copepods were used per treatment; males or females; gravid or non-gravid adults?

Answer: We restated. “As in the previous study of Park et al. (2023), direct and indirect toxin induction were tested simultaneously using experimental cages which are 1 L polycarbonate beakers with nested cages (500 ml) whose bottoms consisted of 10 µm mesh. In brief, A. catenella (initial cell concentration: 300 cells ml−1) within the cage experienced direct toxin induction with 18 adult-female A. hudsonica, whereas cells below the mesh experienced indirect induction (predator cues and the release of prey cellular contents via sloppy feeding). Similar caged containers with no predators served as controls (Assay 2, Fig. 6A).” (lines 406-413). Copepods were starved in 0.2 µm filtered seawater for 24 h to ensure grazers had completely voided their guts (Dam and Peterson, 1988). While all copepods were female stage C6, we could not be certain that all of them were gravid females.

In the direct predation treatments what percentage of the dinoflagellates were consumed?

Answer: 68%. The grazing rate (68%) and fitness cost of toxin production (32%) in the direct predation treatments were derived from Equation 2.6 and 2.7 (see Fig. 3c) in Park and Dam 2021 using the ratio of relative gene expression of the cyc gene, a marker of cell growth.

Thank you.

Reviewer 3 Report

Comments and Suggestions for Authors

The manuscript presents an interesting objective in relation to the interaction of past predator exposure and resource availability on toxin retention and cell growth in  dinoflagellate Alexandrium catenella. I consider the authors' work efficient in developing the experiments and presenting the results with clear graphics. I only have a few questions and suggestions, which I list below:

Line 16: Some words, have been joined together, but I consider it a writing  mistake.

Line 23: I suggest reducing keywords without long phrases

Line 59: STX, the acronym should be clear the first time it is mentioned, it is only clear in Math and Meth.

Lines 127, 154, 405: I suggest clarifying  NRPR1, 2, 3 etc. in the caption of the graphs and table.

Line 366: Why were the copepods previously fed with diatoms and chlorophytes and not with non-toxic dinoflagellates?, please clarify why those species in particular.

Line 452 : I suggest clarifying how the nutrients P and N were measured and the statistics for them.

Author Response

We thank the reviewer for their thoughtful concerns, comments, and suggestions, which we address below. Reviewer’s text is in black, and our answers are in red.

The manuscript presents an interesting objective in relation to the interaction of past predator exposure and resource availability on toxin retention and cell growth in dinoflagellate Alexandrium catenella. I consider the authors' work efficient in developing the experiments and presenting the results with clear graphics. I only have a few questions and suggestions, which I list below:

Line 16: Some words, have been joined together, but I consider it a writing  mistake.

Answer: Fixed. (line 16)

Line 23: I suggest reducing keywords without long phrases

Alexandrium catenella; dinoflagellate; Paralytic Shellfish Toxin (PST); predator-induced toxin production; predator legacy effects on prey; resource-mediated non-consumptive effects; cost of toxin maintenance; defense-growth tradeoff; harmful algal bloom; toxic prey.

Answer: Reduced. (lines 23-28): Alexandrium catenella; Paralytic Shellfish Toxin; past predator-induced toxin; predator legacy effects; non-consumptive effects; toxin retention; toxin maintenance cost; defense-growth tradeoff. See lines 22-24.

Line 59: STX, the acronym should be clear the first time it is mentioned, it is only clear in Math and Meth.

Answer: Saxitoxin (STX) (line 450)

Lines 127, 154, 405: I suggest clarifying  NRPR1, 2, 3 etc. in the caption of the graphs and table.

Answer: We clarified the nutrient regime in the caption of the graphs and table.

Line 366: Why were the copepods previously fed with diatoms and chlorophytes and not with non-toxic dinoflagellates?, please clarify why those species in particular.

Answer: Species of the dinoflagellate genus Alexandrium are well known for their chemical defense traits. However, non-toxigenic Alexandriumspecies can display other chemical compound such as bioluminescence, bioactive extracellular compounds including reactive oxygen species, and allelopathic substances (Granéli et al. 2008; Flores et al. 2012; Tillmann et al. 2020; Long et al. 2021). Leitão et al. (2024) reported the antagonistic effects of three strains of the dinoflagellate HAB species A. catenella (2 toxigenic strain and 1 negligible toxic strain) on two target species (the chlorophyte Tetraselmis sp. and the cryptomonad Rhodomonas salina) under ocean warming and acidification conditions, and interestingly A. catenella showed a negative growth rate in the presence of the diatom Thalassiosira weissflogii.

Granéli, E., Salomon, P.S., & Fistarol, G.O. (2008). The role of allelopathy for harmful algae bloom formation. Algal Toxins: Nature, Occurrence, Effect and Detection, V. Evangelista, L. Barsanti, A.M. Frassanito, V. Passarelli, & P. Gualtieri (eds.); pp.159–178. Springer, The Netherlands.

Flores HS, Wikfors GH, Dam HG. Reactive oxygen species are linked to the toxicity of the dinof lagellate Alexandrium spp. to protists. Aquat Microb Ecol 2012;66:199–209.

Tillmann U, Krock B, Wietkamp S et al. A Mediterranean Alexandrium taylorii (Dinophyceae) strain produces goniodomin A and lytic compounds but not paralytic shellfish toxins. Toxins 2020;2020:564.

Long M, Krock B, Castrec J et al. Unknown extracellular and bioactive metabolites of the genus Alexandrium: a review of overlooked toxins. Toxins 2021;13:905.

Leitão E, Castellanos DF, Park G, Dam HG. Antagonistic interactions of the dinoflagellate Alexandrium catenella under simultaneous warming and acidification. Harmful Algae. 2024. Apr 1;134:102625.

Line 452 : I suggest clarifying how the nutrients P and N were measured and the statistics for them.

Answer: Unfortunately, we did not analyze nutrients at the end of assays, but we measured initial N and P (3 µM L-1 [NO3-]; 1 µM L-1 [PO43-]) using SmartChem nutrient analyzer to set up exact media concentrations. (lines 416-418)

Thank you.

Reviewer 4 Report

Comments and Suggestions for Authors

I have reviewed the manuscript entitled “The ghost of predator past: interaction of past predator exposure and resource availability on toxin retention and cell growth in a dinoflagellate” submitted to Toxins. The manuscript investigates the combined effects of direct and indirect exposure and nutrient-replete and -deplete conditions on toxin production by the dinoflagellate Alexandrium catenella. The manuscript is very well written and well presented, addressing a topic of clear interest to the readers of the journal. The experimental design is robust and has been used in previously published studies; the analyses are solid and were properly executed. As a single minor comment, I would suggest splitting Section 5.4 into two separate sections, one dedicated specifically to the methodology for toxin quantification. However, I acknowledge that this is more a matter of personal preference than a scientific necessity. Thus, I feel comfortable recommending the publication of the manuscript in its current form. Congratulations to the co-authors involved in the execution, writing, and supervision of this study.

Author Response

We thank the reviewer for their thoughtful concerns, comments, and suggestions, which we address below. Reviewer’s text is in black, and our answers are in red.

I have reviewed the manuscript entitled “The ghost of predator past: interaction of past predator exposure and resource availability on toxin retention and cell growth in a dinoflagellate” submitted to Toxins. The manuscript investigates the combined effects of direct and indirect exposure and nutrient-replete and -deplete conditions on toxin production by the dinoflagellate Alexandrium catenella. The manuscript is very well written and well presented, addressing a topic of clear interest to the readers of the journal. The experimental design is robust and has been used in previously published studies; the analyses are solid and were properly executed. As a single minor comment, I would suggest splitting Section 5.4 into two separate sections, one dedicated specifically to the methodology for toxin quantification. However, I acknowledge that this is more a matter of personal preference than a scientific necessity. Thus, I feel comfortable recommending the publication of the manuscript in its current form. Congratulations to the co-authors involved in the execution, writing, and supervision of this study.

Answer: We appreciate your suggestion. The Section 5.4 was divided into two separate sections and reordered in reflection of review’s intention (lines 401-457). Please find the attached. Thank you.